# Effect of Crosshead Speed and Volume Ratio on Compressive Mechanical Properties of Mono- and Double-Gyroid Structures Made of Inconel 718

**DOI:** 10.3390/ma16144973

**Published:** 2023-07-12

**Authors:** Katarina Monkova, Peter Pavol Monka, George A. Pantazopoulos, Anagnostis I. Toulfatzis, Anna Šmeringaiová, Jozef Török, Sofia Papadopoulou

**Affiliations:** 1Faculty of Manufacturing Technologies, Technical University in Kosice, Sturova 31, 080 01 Presov, Slovakia; peter.pavol.monka@tuke.sk (P.P.M.); anna.smeringaiova@tuke.sk (A.Š.); jozef.torok@tuke.sk (J.T.); 2Faculty of Technology, Tomas Bata University in Zlin, Nam. T.G. Masaryka 275, 760 01 Zlin, Czech Republic; 3ELKEME Hellenic Research Centre for Metals S.A., 61st km Athens—Lamia National Road, 32011 Oinofyta, Greece; atoulfatzis@elkeme.vionet.gr (A.I.T.); spapadopoulou@elkeme.gr (S.P.)

**Keywords:** compression properties, cellular structure, additive manufacturing, volume ratio

## Abstract

The current development of additive technologies brings not only new possibilities but also new challenges. One of them is the use of regular cellular materials in various components and constructions so that they fully utilize the potential of porous structures and their advantages related to weight reduction and material-saving while maintaining the required safety and operational reliability of devices containing such components. It is therefore very important to know the properties of such materials and their behavior under different types of loads. The article deals with the investigation of the mechanical properties of porous structures made by the Direct Metal Laser Sintering (DMLS) of Inconel 718. Two types of basic cell topology, mono-structure Gyroid (G) and double-structure Gyroid + Gyroid (GG), with material volume ratios of 10, 15 and 20 %, were studied within our research to compare their properties under quasi-static compressive loading. The testing procedure was performed at ambient temperature with a servo-hydraulic testing machine at three different crosshead testing speeds. The recorded data were processed, while the stress–strain curves were plotted, and Young’s modulus, the yield strength Re_0.2_, and the stress at the first peak of the local maximum *σ*_LocMax_ were identified. The results showed the best behavior under compression load among the studied structures displayed by mono-structure Gyroid at 10 %. At the same time, it can be concluded that the wall thickness of the structure plays an important role in the compressive properties but on the other hand, crosshead speed doesn´t influence results significantly.

## 1. Introduction

The research and creation of new materials is an important area for the improvement of technologies and products. Ultimately, materials with exceptional properties applicable to practice are sought. Such materials, which can have exceptional properties in combination with their weight, can be considered cellular materials.

People have been using cell-structured bodies for centuries, as they are also found in nature (for example, wood, cork, mushrooms, corals, bones, stems and leaves of plants, etc.) and are the building materials for many products. They began to be artificially produced at the beginning of the 20th century when the first commercial product was a rubber sponge. Subsequently, the expansion of the production of cellular bodies from polymers began, and nowadays, cellular bodies are mainly used as thermal insulation, packaging, and protective materials, but also as construction materials mainly due to their strong ability to absorb dynamic shocks, also to absorb adverse sound effects [1,2,3]. Their mechanical properties are not negligible due to their weight, which makes them an ideal material suitable for use in the space, aviation, or automotive industries [4,5].

Research has shown that the behavior and properties of bodies with uniformly distributed cells are much more predictable than those of porous foams in which the cells are randomly distributed in the volume of the body (in terms of their topology and also their size, even if their size is within a certain set range) [6]. However, the production of components with a precisely repeating structure, distributed radially or axially (in one to three axes) was very problematic for structural materials until recently. This problem has been eliminated by 3D printing technology, the biggest advantage of which is the possibility of manufacturing components with complex shapes, which also include lightweight porous materials with a precisely defined cell distribution [7].

The properties of cellular lightweight bodies are the result of a combination of the properties of the cellular structure and the properties of the material used for their production [8]. These two factors together with the volume ratio of the cell bodies that is given by Equation (1) [8]
*Vr* = (*material volume*)/(*total sample volume*),(1)
are the determining parameters for their physical and mechanical properties.

According to Mahmoud et al. [9], the mechanical properties of lightweight structures are influenced not only by design factors but also by production. Due to the fact that for metallic components produced by Direct Metal Laser Sintering (DMLS) technology, the possibilities of investigating the influence of technological factors are limited for the institutions that have acquired the production equipment (as they are given by the manufacturer), most authors focus on investigating the influence of factors related to design (such as volume fraction, porosity, pore size, cell type, and others). However, due to the affordability of plastic materials, as well as the availability of 3D printers for plastic materials, studies aimed at investigating the mechanical properties of structures made of plastic have been carried out to date far more than those in which metal was used for the production of structures [10,11,12,13,14].

Specific mechanical properties exhibit cell structures with a negative Poisson’s ratio. Different properties compared to conventional metamaterials are caused by the auxetic effect [15,16].

The theoretical basis for the description of the mechanical behavior of porous cellular structures under compression was developed by Gibson et al. [17]. Based on their model, the relative elastic modulus of a structure is defined by Equation (2)
*E** = *E_L_*/*E_S_*(2)
where: -*E_L_* is the elastic modulus of the lattice structure;-*E_S_* is the elastic modulus of the solid material.

Both values of elastic modulus are connected with Equation (3), often used to correlate stiffness:*E** = *C*_1_*ρ**^*n*^(3)
where:-*C*_1_ is the coefficient as the range of values from 0.1 to 4.0;-*ρ** is a ratio of the density of the lattice structure *ρ_L_* to the density of the base solid material *ρ_S_*;-*n* is constant with values of approximately 2.

The Gibson–Ashby [17] model uses Equation (4) to correlate stiffness and yield strength with density in highly porous structures, as well as define the association between relative density and densification strain *ε_D_* given by Equation (5):*σ_L_* = *C*_2_*ρ***^m^σ_S_*
(4)
*ε_D_* = 1 − *αρ*(5)
where:-*σ_L_* is the yield strength of the lattice structure;-*σ_S_* is the yield strength of the base solid (pore-free) material;-*C*_2_ is the coefficient of the range of values from 0.25 to 0.35;-*m* is a constant (value about 3/2);-*α* is dependent on the behavior of the matrix material deformation.

So, it can be expressed that coefficients *C*_1_ and *C*_2_ are constants, which depend on the material and on the cell architecture, and *n* and *m* are two other constants depending on the configuration of struts relative to the loading direction [18].

In particular, where bending dominates the deformation behavior, *C*_1_ = 1 and *n* = 2, while for foams where deformation occurs via stretching, *C*_1_ = 0.3 and *n* = 1 [19,20].

Kadkhodapour et al. [21] reported 0.904 and 1.658 for the exponent of the elastic Young’s modulus for cubic and diamond structures that mainly deform by stretching and bending, respectively. Yan et al. [22] investigated the elastic modulus of Gyroid and Diamond structures with various densities and reported *C*_1_ = 0.19 and 0.17, respectively, and *n* = 1.71 and 1.64, respectively. They also considered yield strength, reporting *C*_2_ = 1.31 and 1.39, respectively, and *m* = 1.83 and 1.95, respectively [18].

The real value of the constant parameters *C*_1_, *C*_2_, *n*, *m*, and α are calculated based on the results of the compression test [23,24].

There are many types of periodically cell-patterned structures—from simple cubic cell design, through to honeycomb, and to structures based on so-called Triply Periodic Minimal Surfaces (TPMS). These are surfaces that have a crystalline structure repeating in three directions and whose basis is a cell with a surface topology that locally minimizes the area. Such minimal surfaces necessarily have zero mean curvature; i.e., the sum of the principal curvatures at each point is zero [25].

Structures with complex geometry within so-called Triply Periodic Minimal Surfaces (TPMSs) have become a subject of research for their extraordinary mechanical and damping properties in terms of component weight and material consumption for their production [26,27,28].

A unique place among these TPMSs is the Gyroid, the geometry of which is defined by Equation (6): [29]
sin *x* cos *y* + sin *y* cos *z +* sin *z* cos *x =* 0(6)

For example, Nazir et al. studied in both ways, experimentally as well as numerically, a novel surface-based structure named O-surface that is designed and inspired by existing Triply Periodic Minimal Surface morphologies in a particular sea urchin structure and made by multi-jet fusion technology. To compare the mechanical performance of the structures in terms of compression, local buckling, global buckling, and post-buckling behavior, they were designed with two different height configurations. Their results showed that the sea urchin structure exhibited better mechanical strength than its counterpart, with the same relative density, almost two-fold higher in the compressive response [30].

The Gyroid separates space into two oppositely congruent labyrinths of passages. Channels run through the Gyroid labyrinths in the (100) and (111) directions; passages emerge at 70.5-degree angles to any given channel as it is traversed, the direction at which they do so gyrating down the channel, giving rise to the name “Gyroid”. In 2017, MIT researchers studied the possibility of using the Gyroid shape to turn bi-dimensional materials, such as graphene, into a three-dimensional structural material with low density, yet high tensile strength [31].

Gyroid structures along with diamond ones fabricated using electron beam melting (EBM) technology from Ti-6Al-4V alloy were investigated by Yanez et al. [32] who observed that the specific strength (compressive strength versus density) for Gyroid structures is correlated with the wall angle. By reducing the angle, Young´s modulus and the compressive ultimate strength limit increased. Furthermore, they concluded that Gyroid structures exhibited better strength-to-weight ratios compared to other TPMS structures.

Bobber et al. [33] assessed three types of periodic structures, Primitive, Diamond, and Gyroid, produced from Ti-6Al-4V alloy by means of selective laser melting technology. For each unit cell, the porosity and surface area decrease with increasing wall thickness. The investigated specimens also showed outstanding high fatigue resistance and an exceptional combination of a relatively low modulus of elasticity and a high yield strength. One form of the Gyroid lattice known as the double-Gyroid was recently identified as having high stiffness and low maximum von Mises stress compared to a variety of other cell types, making it particularly suitable for use in lightweight components. Furthermore, Aremu et al. [34] noted that the double-Gyroid lattice, unlike several other lattice types, possesses axisymmetric stiffness, again making it a good candidate for applications where the exact nature and direction of the loads are not fully known or if they are subject to large uncertainties.

The energy absorption of as-built and heat-treated double Gyroid lattice structures was studied by Maskery et al. [35]. Their results showed that the total energy absorbed by heat-treated double-Gyroid lattices up to 50 % was close to three times the energy absorbed by the examined single BCC lattices, which had the same volume ratio.

Scientists and engineers are interested in Gyroids because of the way they interact with both light and sound waves, promising nanoscale materials with novel properties. The Gyroid’s form dictates how and even whether a wave will pass through to the other side. In that way, the material can be invisible to some waves or a reflector of other wavelengths. Remarkably, the chemical combination of polydimethylsiloxane (PDMS) and polystyrene, initially dissolved in a solution, self-assembles into a double Gyroid, with two distinct PDMS networks dancing around each other without ever touching. A double Gyroid can be even more tunable, as distinct materials making each network could affect signals differently. All this is predicated on the unit cell structure being perfect cubes [36].

The properties of double Gyroids, that the constituent interpenetrating networks are single Gyroids of opposite chirality, were also studied by Indurkar et al. They stated that it is straightforward to induce connectivity between the two networks by translating one with respect to the other. The interconnected and interpenetrating Gyroids of opposite chirality are a bending-dominated topology. The increase in nodal connectivity via the inducing interconnectivity within them is insufficient to switch their behavior to stretching-dominated [37].

The mechanical properties of mono-Gyroid structures (of different types—Sheet-based and Strut-based Gyroids [38]) made of different materials with different densities have been investigated in many studies. However, from the point of view of mechanical properties, the double Gyroid has remained in the background of scientific research until Gyroid + Gyroid is the name of a double cell structure now, despite the fact that its other properties (especially optical, photonic, electrical, sound, and others) are relatively well-researched, an example of which is the comprehensive study of Sherer [39].

Since other types of properties of the double Gyroid look advantageous and promising for their application, the authors, therefore, decided to devote themselves to the study of this double-cell structure Gyroid—Gyroid also within the research of the influence of cross-head speed on its properties in comparison with the properties of a mono Gyroid.

Within the presented research, two types of cell-structured samples were examined, i.e., one sample was designed as a basic simple mono-structure Gyroid (G) and one as a double-structure Gyroid + Gyroid (GG), which was formed by the mutual displacement of cells of the same type while keeping the same cell sizes of 10 × 10 × 10 mm in all samples and total volume ratios of 10, 15, and 20 % (± 0.5 %). The aim of the paper was therefore not only to compare the mechanical properties of the samples under pressure with three different crosshead speeds of 1, 10, and 100 mm/min, but also to specify which type of sample is more suitable for compressive loading and whether it is a basic mono-structured or double-structured one while maintaining the cell size and total dimensions of the samples as the comparison criteria.

A few research studies [40,41,42,43] have already been conducted to study the effect of cross-head speed on compressive mechanical properties; however, to the best of the authors’ knowledge, Gyroid structures, arranged in the way described above, made from Inconel 718, have not been investigated so far, while at the same time, the influence of cross-head velocity on the properties of these structures under compressive loading has not been investigated. Therefore, the authors consider the results achieved as a novelty and their contribution to this field.

## 2. Materials and Methods

### 2.1. Characteristics of the Investigated Samples

Two types of structure were selected for testing within the research, namely, Gyroid (G) and Gyroid + Gyroid (GG) with volume ratios of *Vr* = 10, 15, and 20 %. The specimens were produced from EOS nickel alloy IN718 (Inconel 718) powder with a generic particle size distribution of 20–55 µm, which chemical composition, listed in Table 1, corresponding to UNS N07718, AMS 5662, AMS 5664, W.Nr 2.4668, DIN NiCr19Fe19NbMo3 [44,45].

Inconel 718 is a nickel-based alloy with exceptional properties and technical specifications.

This alloy also exhibits good weldability, which makes it one of the most versatile materials on the market today.

One of the outstanding features of Inconel 718 is that it is tremendously versatile and easy to work with. It is significantly better than other nickel-based alloys, especially those hardened with aluminum or titanium. Its biggest advantage over other construction materials is the combination of excellent mechanical properties with great resistance against corrosion and oxidation at high temperatures. Moreover, when comparing the materials in terms of performance, Inconel 718 is renowned for its excellent yield, tensile, impact, and fatigue strength, creep resistance, thermal stability, and resistance to stress cracking in chloride solutions. Additionally, its low coefficient of thermal expansion allows components made from this material to maintain their integrity during rapid changes in temperature. It can be used in environments ranging from cryogenic temperatures, where components need to maintain structural integrity even when exposed to extremely cold temperatures, up to 700 °C. Because the alloy can be fabricated so easily and into increasingly complex parts, the material is suitable for high-performance applications, making it the best choice for aircraft parts such as turbine blades and other components exposed to extreme stress or heat. Engineers are constantly striving to improve the reliability and efficiency of engines, not to mention their longevity, so nickel alloy 718 is therefore chosen for its balance between cost and performance [47,48,49,50,51].

Virtual 3D models of the samples with overall dimensions of 40 × 40 × 50 mm were prepared in PTC Creo 8 software. A Gyroid (G) basic cell with dimensions of 10 × 10 × 10 mm was regularly patterned in three orthogonal directions (in the *x*-, *y*-, and *z*-axes), whereas, in the double-structure Gyroid + Gyroid (GG), individual types of cells were moved relative to each other by 5 mm in each direction, as explained in Figure 1. Thus, these combined double structures were arranged twice as densely, but to maintain the same volume ratio as in the “simple” structures, the structure’s wall was thinner (half).

Briefly, 3D-printed Inconel 718 cellular specimens were manufactured by Direct Metal Laser Sintering (DMLS) technology employing EOS EOSINT M290 (EOS GmbH, Munich, Germany). Because of problems with the structure overhanging in the corners, difficulties with accurately manufacturing the samples according to the 3D models appeared, so the corners were supported as shown in Figure 2. The samples were joined to the building platform with another support structure, and since the Gyroid is a self-supporting structure, no more supports were needed.

All the supporting structures were cut off from the specimens by applying an electric discharge machine at a hand workshop after heat treatment procedure AMS 5664 according to the EOS Inconel 718 datasheet [44], which was realized in the following steps: solution annealing at 1065 °C for 1 h, cooling in argon gas, aging treatment held at 760 °C for 10 h, furnace cooling to 650 °C for 2 h, held at 650 °C for 8 h, followed by Argon gas cooling. The final post-processing procedure consisted of grinding to the final sizes followed by fine sandblasting. The produced samples are presented in Figure 3.

Due to the high production price of samples made of Inconel 718 and due to the limited possibilities of the authors’ workplaces, only one piece per order of each type of sample was produced in the company Innomia, a.s. (Jaromeř, Czech Republic), the quality of which was thoroughly and consistently checked at the output. This company has been involved in the production of components using DMLS technology for a long time and has extensive experience in this field.

### 2.2. Testing Procedure

Compression tests were performed according to the ASTM E9 international standard at ambient temperature, employing a servo-hydraulic testing machine Instron 8802 with a maximum capacity of 250 kN (Figure 4).

The testing procedure was realized using position mode under different crosshead speeds: 1 mm/min, 10 mm/min, and 100 mm/min. So, a total of eighteen (18) pieces of 3D-printed Inconel 718 cellular specimens with codification, as shown in Table 2, were tested to find out the effects of crosshead speed and the suitability of mono- or double-structured cell types on compressive mechanical properties.

### 2.3. Evaluation of Energy Absorption

The behavior of materials and structures plays a key role in the safety of products under load. Energy absorption refers to the process of dissipating input energy from an external load through plastic deformation or fracture and it corresponds to the action of the force on the displacement [52]. A force *F* acting through an elementary displacement d*L* performs elementary work *F*d*L*. A stress *σ* = *F/A* acting through a strain increment d*ε* = d*L/L* performs work:d*W* = *F*d*L*/*AL* = *σ* d*ε*
(7)
where *A* is a cross-sectional area on which the force acts. If the stress is acting on an elastic material, this work is stored as elastic energy. The work performed per unit volume as the stress is raised from zero to a final value *σ** is the area under the stress–strain curve:
(8)$$W=\int_{0}^{\sigma */E}\sigma d\varepsilon =\frac{1}{2}\frac{{\left(\sigma *\right)}^{2}}{E}$$

This is the energy *W* (J) that is stored, per unit volume, in an elastically strained material. The energy is released when the stress is relaxed.

Within our research, the measured force-displacement curves were used for calculating the energy absorptions of the individual samples (differing in the volume ratio *Vr*, type of structure, G and GG, as well as crosshead speed). Although the compression tests were carried out without splintering fracture up to approx. 80 % compressive strain, the final strains varied. In order to make the results comparable, the authors decided to use a strain of 60 % as a criterion for evaluating energy absorption.

Because of the plateau regions were not smooth, it was difficult to describe the force and displacement dependencies using equations. This was the reason why the authors decided to use the discreditation method to calculate the energy instead of differentiation. The principle (Figure 5) consists of dividing a large area into the sum of small areas corresponding to the area of the rectangle with sizes Δ*a* and height *f*(*a*), where Δ*a* is the increment of the value on the *x*-axis and *f*(*a*) is the functional value corresponding to the variable *a*, which hardly changes in the interval Δ*a* [53]. The software MS Excel 2021 was used for the area (i.e., energy absorption) calculation.

## 3. Results and Discussion

Data from the experimental process were recorded. The curves of the dependence of compression load force on displacement for all the investigated types of structure and for all three crosshead speeds were plotted. An example of the dependencies for Gyroid structures with volume ratios of *Vr* = 10, 15, and 20 % at the crosshead speed of 10 mm/min is shown in Figure 6.

Based on the measured data, the values of the first peak local maximum force were identified, and they are listed in Table 3.

Two views of samples with a Gyroid + Gyroid structure after deformation at the crosshead speed of 10 mm/min are in Figure 7. The macrographs in Figure 7 confirmed the deformation capacity of the structures, which possessed rather ductile behavior without displaying destructive fractures.

The measured data were processed, while stress–strain curves were generated in the software application MS Excel taking into account a real cross-section area of each sample that was specified by means of software PTC Creo 8, as shown in Figure 8.

The PTC Creo software uses differential equations to calculate all surface cross-sectional characteristics. The results of the calculations running in the background were analytically calculated and verified by the co-author of this manuscript [53]. An example of the curves of all the studied structures at the crosshead speed of 100 mm/min is presented in Figure 9.

In stress–strain diagrams, the so-called “plateau region” is visible, where the deformations of the cells increase with a minimal change in stress, thereby increasing their ability to absorb energy [54].

Young´s modulus, the yield strength, and the stress at the first peak of the local maximum for all types of structure were identified during processing. Histograms comparing Young’s modulus for both types of structure at all three crosshead speeds are shown in Figure 10.

When comparing Young’s modulus, the graphs above show that for both types of structure, the value of *E* (GPa) decreases with increasing specific mass, while the values for a given specific mass vary only very slightly at different crosshead speeds.

To characterize mono- and double-Gyroid structures at each investigated volume ratio and at each crosshead speed, the values of stress at the first peak of the local maximum *σ*_LocMax_ and yield strength Re_0.2_ were recognized. The achieved results are presented via histograms in Figure 11.

The results show that the crosshead speed in the range of 1–100 mm/min used during testing does not significantly affect the behavior of the structures and their mechanical properties under compression load, especially for 15 % and 20 % volume ratios.

Following the results, it can be stated that the Gyroid + Gyroid (GG) double-structures achieved lower values of the first peak local maximum force load when compared to the Gyroid (G) mono-structures in all three volume ratios and at all three crosshead speeds, while the highest difference was observed in the specimens with volume ratios of 10 %. Thus, it is possible to assume that the thickness of the wall of the structure affects the behavior of the sample to a greater extent than factors related to the testing speeds of the crosshead. One of the other causes of the worse mechanical properties of double-Gyroid structures is the “crossing of the walls”, which can affect not only the technological conditions of production (e.g., laser beam guidance strategy that can thereby influence the microstructure of the material during the passing of the laser in a new layer) but at the same time, it is probably necessary to take into account that at the crossing of the walls, almost sharp inner edges are created, which tend to behave in the sense of a notch, which presumably also subsequently affects the mechanical properties of the double Gyroid.

Based on the values of stresses at the first peak of the local maximum *σ*_LocMax_ and the yield strength Re_0.2_ presented in Figure 11, it can also be seen that specimen G10_1 is superior to others of the same type (G) at higher test speeds and also higher than all types of GG structures at all speeds. The reason for this result can probably be found in the appropriate combination of the thickness of the cell wall of the structure, the cell side length, and the speed of the crossbar, which resulted in the densification of the structure G10_1 as a result of the beginning of the interaction of the walls in the cell, which increased the pressure resistance of the cellular solid [55]. Currently, there is ambiguity in the definition and uncertainty in determining the compressive strain at which the densification regime begins. The initial densification strain was defined and distinguished in the studies of Li [56] and Wang [57], where the influence of geometric design parameters, including cell size length and wall thickness on the onset strain of densification, as well as the calculation formula between the onset strain of densification and geometric design parameters, have been derived.

At the same time, it can be assumed that the setting parameters within the production conditions were the most suitable for the sample with this volume ratio and that the individual layers of the material were best connected each to other during sintering.

Since the effect of the crosshead speed on the compressive properties of the samples has not been observed, the authors also decided to look at the samples´ behavior from the point of view of energy absorption. Based on the methodology described in Section 2.3, the dependencies of the energy absorption at 60 % strain on both the crosshead speed and the volume ratio were plotted separately for Gyroid and Gyroid + Gyroid structures. They are presented in Figure 12.

It is also visible from the graphs above that in this case, the crosshead speed did not influence the amount of energy, while with the volume ratio, the amount of energy increased in the Gyroid in an almost linear way and in the Gyroid + Gyroid in a slightly exponential way. Comparing the maximal values, it can be said that a slightly higher result in energy absorption (6012.047 J) was achieved by the Gyroid structure, while with the Gyroid + Gyroid structure, it was 5955.365 J, both at a crosshead speed of 10 mm/min.

Although the results did not show better compressive properties of the Gyroid + Gyroid structure, the porous structure of the Gyroid itself is characterized by a large surface area, which is very useful for heat conduction and light absorption (or the accumulation of light energy). This function is doubled by the Gyroid + Gyroid structure, so the use of the double-Gyroid structure is promising for heat exchangers and batteries, increasing their capacity.

Although the results do not show the better compression properties of Gyroid + Gyroid type of structure, the Gyroid’s porous structure is characterized by a large surface area, which is very useful for heat conduction and light absorption (or light energy storage). This feature is doubled by the Gyroid + Gyroid structure, so the use of the double Gyroid structure is promising for heat exchangers and batteries, thereby increasing their capacity.

## 4. Conclusions

In nature, there are many types of materials with porous structures, which have become an inspiration for the development of new materials used in various sectors of industry, economy, and everyday use. Their properties usually bring benefits not only from the point of view of saving materials in relation to their physical or mechanical properties, but their application in practice brings secondary benefits with respect to the environment (for example, when the weight of an airplane or car is reduced, the amount of fuel consumed is reduced and thus also the fumes escaping into the environment, etc.). Each of the porous structures, whether natural or artificially made, exhibits different properties. For a designer who wants to use this type of lightweight material in a design, it is, therefore, advantageous to know the behavior of individual types of structure and to be able to choose the one that is most suitable from the point of view of the expected stress on the component (product).

As part of the presented research, two types of Gyroid structures were experimentally tested, namely mono- (G) and double (GG)- Gyroid structures, while the influence of crosshead speed (at three different values of 1, 10, and 100 mm/min) on the compression properties of the samples were studied. Samples were made from Inconel 718 by DMLS technology in three different volume ratios of 10, 15, and 20 %.

The measured values of the first peak of the local maximum force varied for the samples with the highest volume ratio of 20 % in the range of 225–228 kN for the mono-Gyroid and in the range of 219–224 kN for the double-Gyroid. At the smallest volume ratio *Vr* = 10 %, the measured maximum forces reached values of 99–109 kN for the mono Gyroid and 85–91 for the double Gyroid.

The highest Young’s modulus of elasticity *E* = 58 GPa was identified for sample G10_1 and the lowest of 33 GPa for sample G20_3. The G10_1 sample also showed the best other compression properties (yield strength Re_0.2_ = 874 MPa and stress at the first peak of local maximum *σ*_LocMax_ = 1089 MPa), so it can be assumed that there occurred the most suitable combination of the parameters related to the production conditions, the cell wall thickness of the structure, the cell size, and the loading speed, which resulted in densification and increased the pressure resistance of the cellular solid.

The results showed that the speed of the crosshead (in the used range of 1–100 mm/min) does not affect the behavior of the structures under pressure loading.

The amount of energy absorption increases with the volume ratio *Vr* and it does not differ very much when comparing both types of structure, G and GG.

Another important finding that emerged from the research is that the double structure Gyroid + Gyroid at the same volume ratio of materials shows worse properties compared to the mono structure, which is probably related not only to the lower thickness of the wall but also to the crossing of the walls, which changes the technological conditions and at the same time creates weaker places in the form of internal corners that behave as notches.

## Figures and Tables

**Figure 1 materials-16-04973-f001:**
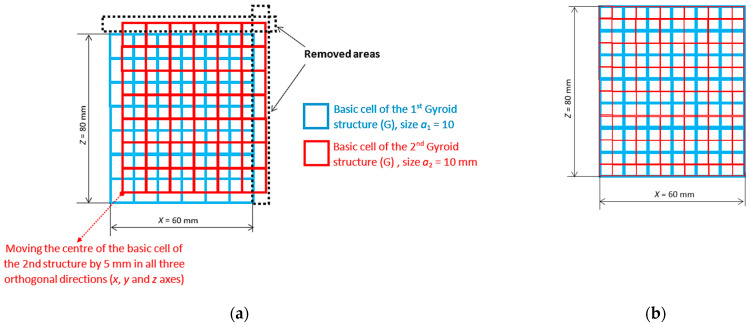
Arrangement of cells in the double-structure Gyroid + Gyroid, (**a**) principle of the double structure GG creation, (**b**) resulting simplified scheme of the double structure GG.

**Figure 2 materials-16-04973-f002:**
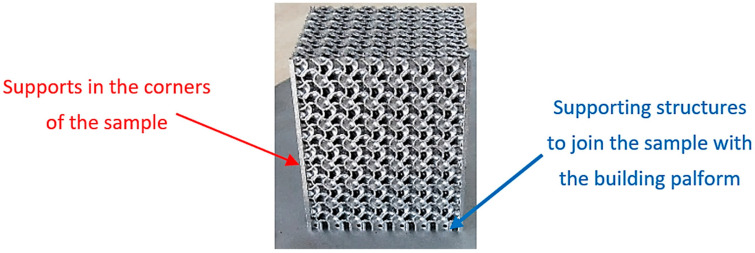
Support structures used during the sample production.

**Figure 3 materials-16-04973-f003:**
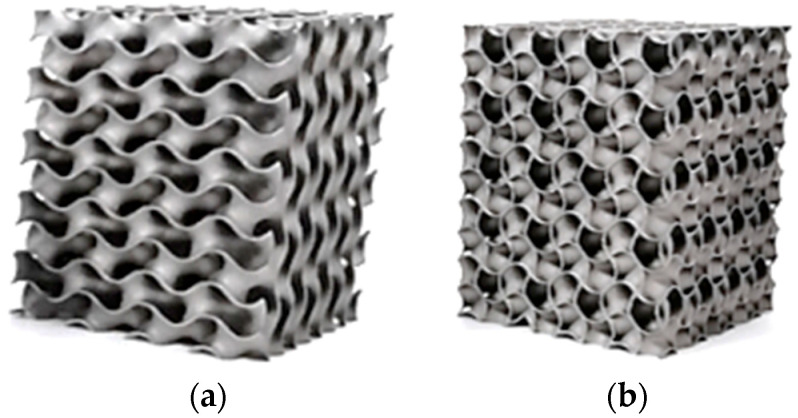
The 3D-printed Inconel 718 cellular specimens with a 10 % volume ratio: (**a**) Gyroid; (**b**) Gyroid + Gyroid.

**Figure 4 materials-16-04973-f004:**
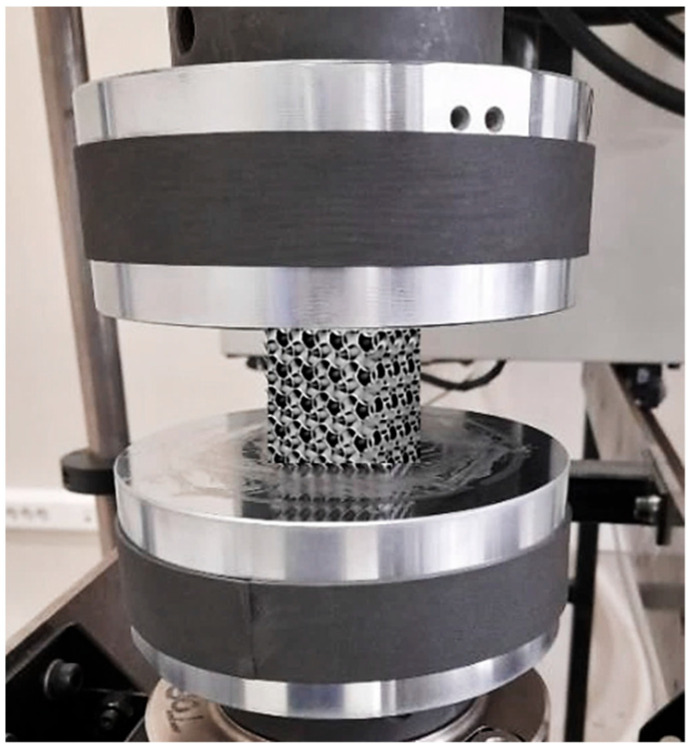
Specimen during the test process under pressure loading.

**Figure 5 materials-16-04973-f005:**
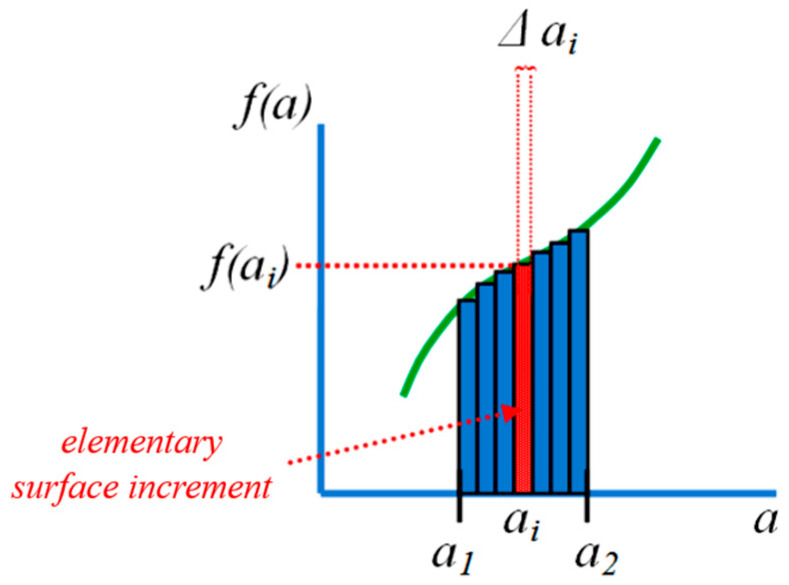
The principle of discreditation at the calculation of energy absorption.

**Figure 6 materials-16-04973-f006:**
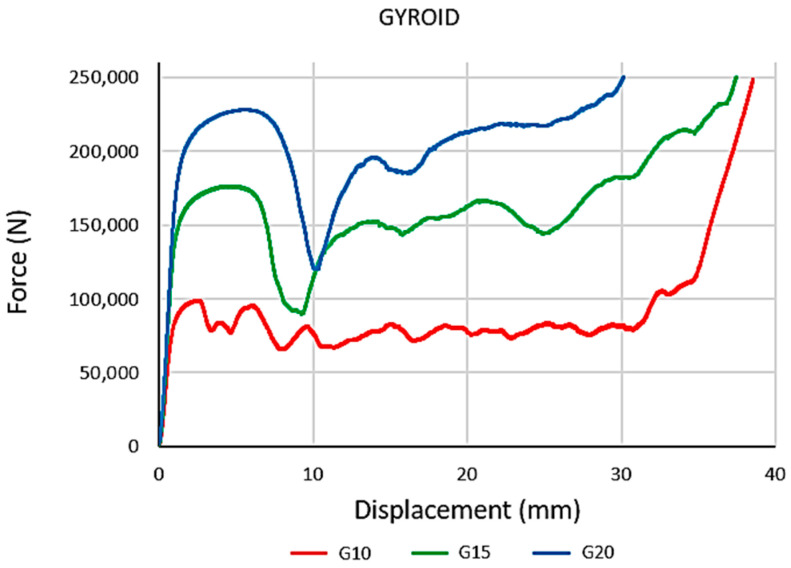
The measured force displacement dependencies for Gyroid structures at crosshead speed 10 mm/min.

**Figure 7 materials-16-04973-f007:**
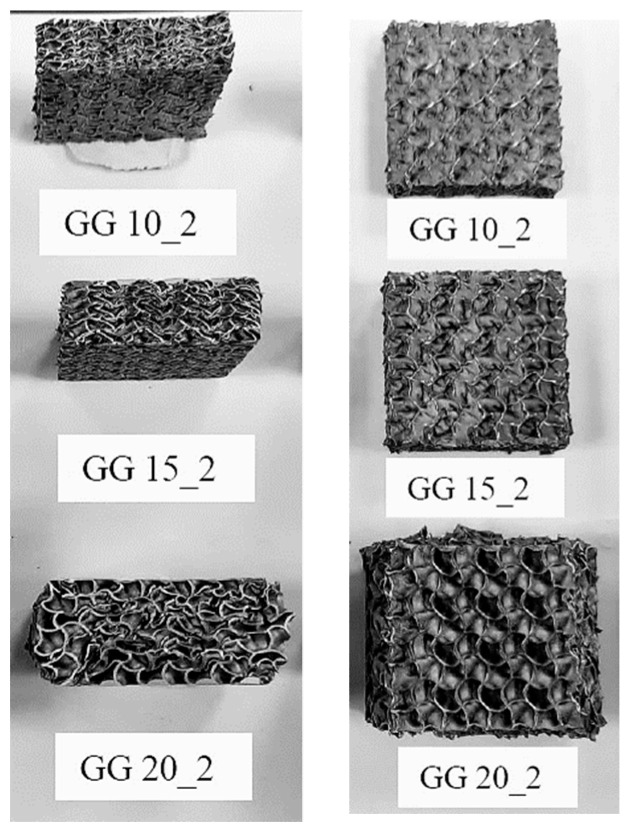
Deformed samples after pressure loading at a 10 mm/min crosshead speed. Compressive displacement was achieved resulting in reaching the maximum load limit without destructive fracture.

**Figure 8 materials-16-04973-f008:**
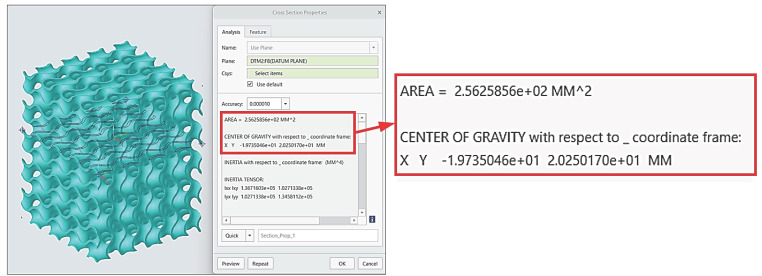
An example of cross-section area specification.

**Figure 9 materials-16-04973-f009:**
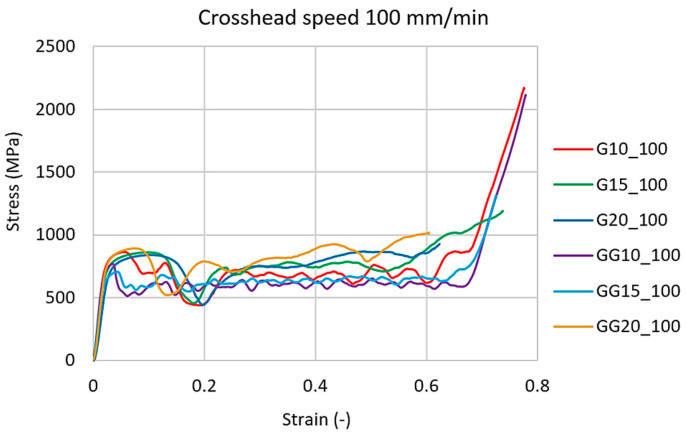
Stress–strain curves for all tested types of structure at a crosshead speed of 100 mm/min.

**Figure 10 materials-16-04973-f010:**
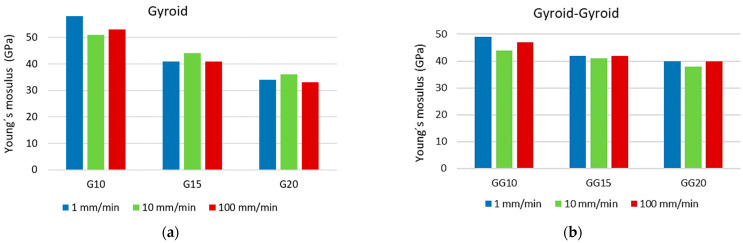
Comparison of Young’s modulus for both types of structure at all three crosshead speeds: (**a**) mono-structure Gyroid; (**b**) double-structure Gyroid + Gyroid.

**Figure 11 materials-16-04973-f011:**
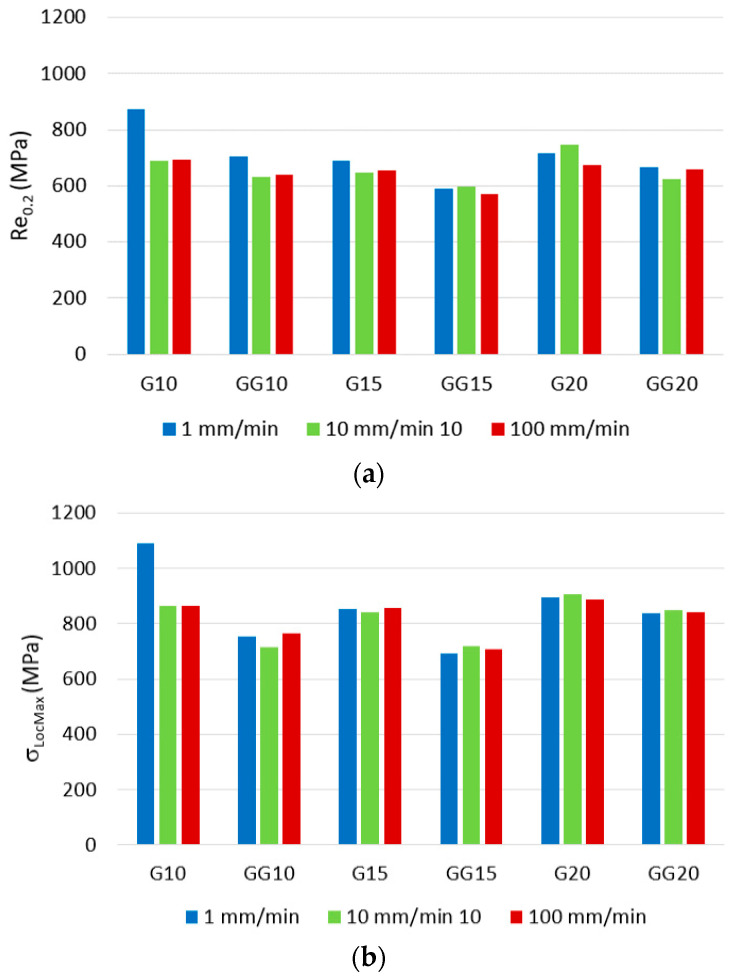
Processed results for both types of structure at all three crosshead speeds: (**a**) yield strength Re_0.2_ of the structures at all three crosshead speeds; (**b**) stress at the first peak of local maximum *σ*_LocMax_.

**Figure 12 materials-16-04973-f012:**
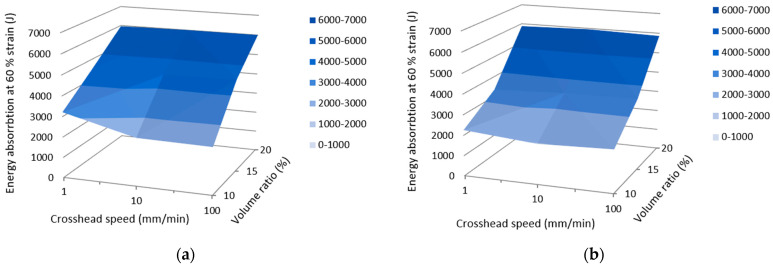
Energy absorption at 60 % of strain: (**a**) Gyroid; (**b**) Gyroid + Gyroid.

**Table 1 materials-16-04973-t001:** Chemical compositions of Inconel 718 (wt. %) [44,46].

Component	Fe	Ni	Cr	Nb	Mo	Ti	Al	Co	Cu	Si	Mn
wt. %	Rem	55	21	5.5	3.3	1.15	0.8	1.0	0.3	0.35	0.35

**Table 2 materials-16-04973-t002:** Specimen codification concerning the structure type, volume ratio, and testing speed of 3D-printed Inconel 718 cellular specimens.

Volume Ratio (%)	Structure Type	Testing Speed		
		1 (mm/min)	10 (mm/min)	100 (mm/min)
10	Gyroid	G10_1	G10_10	G10_100
	Gyroid + Gyroid	GG10_1	GG10_10	GG10_100
15	Gyroid	G15_1	G15_10	G15_100
	Gyroid + Gyroid	GG15_1	GG15_10	GG15_100
20	Gyroid	G20_1	G20_10	G20_100
	Gyroid + Gyroid	GG20_1	GG20_10	GG20_100

**Table 3 materials-16-04973-t003:** Measured values for the First Peak Local Maximum Force (kN).

**Specimen ID**	**First Peak Local Maximum Force (kN)**		
	Crossbar Speed 1 mm/min	Crossbar Speed 10 mm/min	Crossbar Speed 100 mm/min
G10	109	99	99
GG10	89	85	91
G15	179	176	180
GG15	131	137	134
G20	225	228	226
GG20	221	224	219

## Data Availability

Not applicable.

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
