# Peer review of "Effect of Crosshead Speed and Volume Ratio on Compressive Mechanical Properties of Mono- and Double-Gyroid Structures Made of Inconel 718"

_materials, 2023, doi:10.3390/ma16144973_

Round 1

Reviewer 1 Report

1. Explain the meaning of σs.

2. How did the author get the value range of C1 and C2? Please explain the meaning of the parameter.

3. The author uses Inconel718 as the base material. What are the advantages of this material compared with other materials.

4. Whether the conclusion that the performance of G structure is better than that of G+G structure can provide a more adequate explanation.

5. The conclusion is short, and it is suggested that the author revise the conclusion to include more quantitative analysis, rather than just a description of the experimental results.

6. The Introduction of the author needs to be added that the research object of the author is porous materials. Porous materials contain a special kind of porous materials with negative Poisson's ratio properties. Their compression properties and energy absorption properties are different from those of other types of porous materials. The author may refer to the literature:

doi: 10.3390/ma16083262; doi: 10.3390/ma16051808;

Author Response

Dear Reviewer!

Thank You very much for Your valuable comments. We received them with great respect. We also would like to thank You for the possibility to make the improvements. We appreciate it very much.

All changes incorporated in the manuscript are highlighted in blue and responses to You are coloured in green.

  1. Reviewer: Explain the meaning of σs.

The meaning of the σs. was explained below equations (4) and (5).

  1. Reviewer: How did the author get the value range of C1 and C2? Please explain the meaning of the parameter.

As it has been reported in the original manuscript, the real value of the constant parameters C1, C2, n, m, and α are calculated based on the results of the compression test. A few examples of the calculated parameters and closer description of the issue were added in the manuscript within the Introductory section:

“…, it can be expressed that coefficients C1 and C2 are constants, which depend on the material and on the cell architecture, and n and m are two other constants depending on the configuration of struts relative to the loading direction. [18]

In particular, where bending dominates the deformation behavior, C1 = 1 and n = 2, while for foams where deformation occurs by stretching, C1 = 0.3 and n = 1 [19,20].

Kadkhodapour et al. [21] reported 0.904 and 1.658 for the exponent of the elastic Young’s modulus for cubic and diamond structures that mainly deform by stretching and bending, respectively. Yan et al. [22] investigated the elastic modulus of Gyroid and Diamond structures with various densities and report C1 = 0.19 and 0.17, respectively, and n = 1.71 and 1.64, respectively. They also considered yield strength, reporting C2 = 1.31 and 1.39, respectively, and m = 1.83 and 1.95, respectively. [18]”

The authors have not yet calculated the constants and parameters; it will be the next step in their ongoing research as more results are obtained.

  1. Reviewer: The author uses Inconel718 as the base material. What are the advantages of this material compared with other materials.

The properties of Inconel 718 have been rewritten to highlight its advantages over other materials in the way as indicated below. The authors see the greatest advantage of the material over others in the combination of excellent mechanical, physical and other properties of the alloy with its simple fabrication and the subsequent possibility of simple processing even for very complex parts.

One of the outstanding features of Inconel 718 is that it is tremendously versatile and easy to work with. It is significantly better than other nickel-based alloys, especially those hardened with aluminum or titanium. Its biggest advantage over other construction materials is the combination of excellent mechanical properties with great resistance against corrosion and oxidation at high temperatures. More, when comparing the materials in terms of performance, Inconel 718 is renowned for its excellent yield, tensile, impact and fatigue strength, creep resistance, thermal stability, and resistance to stress cracking in chloride solutions. Additionally, its low coefficient of thermal expansion allows components made from this material to maintain their integrity during rapid changes in temperature. It can be used in environments ranging from cryogenic temperatures, where components need to maintain structural integrity even when exposed to extremely cold temperatures, up to 700 °C. Because the alloy can be fabricated so easily and into increasingly complex parts, the material is suitable for high-performance applications, making it the best choice for aircraft parts such as turbine blades and other components exposed to extreme stress or heat. [47-51]

  1. Reviewer: Whether the conclusion that the performance of G structure is better than that of G+G structure can provide a more adequate explanation.

As already mentioned in our article, the mechanical properties of gyroid structures (of different types - Sheet-Based and Strut-Based Gyroid) made of different materials have been investigated in many studies. Despite the fact that its other properties (especially optical, photonic, electrical and others) have been relatively well researched, an example of which is the literature source:

[39] Scherer, M.R.J. (2013). Double-Gyroid-Structured Functional Materials. Springer Theses. doi:10.1007/978-3-319-00354-2,

in terms of mechanical properties, the double gyroid has remained in the background of scientific research so far.

The authors, therefore, decided to devote themselves to the study of this double cell structure Gyroid - Gyroid also within the framework of research on the influence of the speed of the crossbar on its properties when compared to the properties of a simple gyroid. Due to the lack of further research and literary sources, it is therefore not possible to draw definite conclusions yet. On the other hand, not only this, but also our other studies, point out that, from the point of view of mechanical properties, a very important role is played not only by the relative density and topology of the structure as stated in the Gibson-Ashby model, but also by the thickness of the wall of the structure itself (and also the thickness of the shell when applying the structure to the core of the sample/component), which is another parameter that needs to be taken into account in computational and numerical models on which we have already pointed out in the conclusions of the article.

To discuss the reason of better properties of mono Gyroid structure in comparison to double Gyroid, the following text have been added into the manuscript:

“One of the other causes of worse mechanical properties of double Gyroid structure is the "crossing of the walls", which can affect not only the technological conditions of production (e.g. laser beam guidance strategy that thereby can influence the microstructure of the material during the passing of the laser in a new layer) but at the same time, it is probably necessary to take into account that at the crossing of the walls, almost sharp inner edges are created, which tend to behave in the sense of a notch, which presumably also subsequently affects the mechanical properties of the double gyroid.”

  1. Reviewer: The conclusion is short, and it is suggested that the author revise the conclusion to include more quantitative analysis, rather than just a description of the experimental results.

The results have been rewritten in more quantitative way, while the following text has been added into the Conclusions:

The measured values of the First peak local Maximum Force varied for the samples with the highest volume ratio of 20 % in the range of 225-228 kN for the mono Gyroid and in the range of 219-224 kN for the double Gyroid. At the smallest volume ratio Vr = 10 %, the measured Maximum Forces reached values of 99 - 19 kN for the mono Gyroid and 85 - 91 for the double Gyroid.

The highest Young's modulus of elasticity E = 58 GPa was identified for sample G10_1, the lowest 33 GPa for sample G20_3. The G10_1 sample also showed the other best compression properties (yield strength Re0.2 = 874 MPa and stress at the first peak of local maximum sLocMax = 1089 MPa), so it can be assumed that there were occurred the most suitable combination of the parameters related to the production conditions, the cell wall thickness of the structure, cell size and the loading speed that resulted in densification and increased the pressure resistance of the cellular solid.

  1. Reviewer: The Introduction of the author needs to be added that the research object of the author is porous materials. Porous materials contain a special kind of porous materials with negative Poisson's ratio properties. Their compression properties and energy absorption properties are different from those of other types of porous materials. The author may refer to the literature: doi: 10.3390/ma16083262; doi: 10.3390/ma16051808;

There are many types of structures that are divided from different points of view. It was not the aim of the authors to mention all types and compare their mutual properties. We admit that structures with a negative Poison ratio have specific properties, which we also mentioned in the manuscript in the following way:

„Specific mechanical properties exhibit cell structures with a negative Poisson's ratio. Different properties compared to conventional metamaterials are caused by the auxetic effect. [15,16]”

Zhao G, Fu T, Li J. Study on Concave Direction Impact Performance of Similar Concave Hexagon Honeycomb Structure. Materials (Basel). 2023 Apr 21;16(8):3262. doi: 10.3390/ma16083262. PMID: 37110098; PMCID: PMC10147094

Zhao, G.; Fu, T. A Unit Compound Structure Design: Poisson’s Ratio Is Autonomously Adjustable from Negative to Positive. Materials 2023, 16, 1808. https:// doi.org/10.3390/ma16051808

Reviewer 2 Report

Authors have attempted to study the lattice structures mechanical properties at different cross head speeds and found that this parameter doesn’t effect considerably on the performance. Therefore, it shows very less innovative work in this study. Thus, reviewer would like to emphasize to consider some more parameter and their effect on the studied lattices in order to make this study more valuable for the readers and peers. Authors can consider studying the energy absorption behavior of the studied structures as well. Some other comments are as follows.

1.    Research gap is not very strong, authors needs to add some more work as mentioned in comment above. A recent and relevant article is for authors review and consideration. A state of the art review on lattice structures by Nazir et al.

2.    Why double gyroid structure was made? What is the intended application and benefit from this double complexity?

3.    Force and displacement chart are shown in fig. 5. How about stress strain of these structure? Can authors calculate the stress strain from load deformation data and present stress strain graphs accordingly?

4.    It is already well known that cross head speed does not significantly affect the lattice structure property, why is it one of the important parameters in this study. Authors can do a literature review and find out many papers which establish this parameter.

5.    What is the method of calculating area of cross section of gyroid structure shown in fig. 7? Software might not show an appropriate area. For area calculation, authors may review Investigation of torsional properties of surface- and strut-based lattice structures hailu et al. 

English is okay, some minor corrections are needed.

Author Response

Dear Reviewer!

Thank You very much for Your valuable comments. We received them with great respect. We also would like to thank You for the possibility to make the improvements. We appreciate it very much.

All changes incorporated in the manuscript are highlighted in blue and responses to You are coloured in green.

Reviewer: Authors have attempted to study the lattice structures mechanical properties at different cross head speeds and found that this parameter doesn’t affect considerably on the performance. Therefore, it shows very less innovative work in this study. Thus, reviewer would like to emphasize to consider some more parameter and their effect on the studied lattices in order to make this study more valuable for the readers and peers. Authors can consider studying the energy absorption behavior of the studied structures as well. Some other comments are as follows.

  1. Reviewer: Research gap is not very strong, authors needs to add some more work as mentioned in comment above. A recent and relevant article is for authors review and consideration. A state of the art review on lattice structures by Nazir et al.

As part of the introduction, a deeper analysis of the issue was made (we kindly ask the Reviewer to look at the coloured texts in the Introduction section), while the following literary sources were added to the manuscript:

  • Benedetti, M., Klarin, J., Johansson, F., Fontanari, V., Luchin, V., Zappini, G., & Molinari, A. (2019). Study of the Compression Behaviour of Ti6Al4V Trabecular Structures Produced by Additive Laser Manufacturing. Materials, 12(9), 1471. doi:10.3390/ma12091471
  • Deshpande, V.S.; Ashby, M.F.; Fleck, N.A. Foam topology: Bending versus stretching dominated architectures. Acta Mater. 2001, 49, 1035–1040.
  • Wang, A.J.; McDowell, D.L. In-Plane Sti_ness and Yield Strength of Periodic Metal Honeycombs. J. Eng. Mater. Technol. 2004, 126, 137–156.
  • Kadkhodapour, J.; Montazerian, H.; Darabi, A.C.; Anaraki, A.P.; Ahmadi, S.; Zadpoor, A.A.; Schmauder, S. Failure mechanisms of additively manufactured porous biomaterials: E_ects of porosity and type of unit cell, Mech. Behav. Biom. Mater. 2015, 50, 180–191.
  • Yan, C.; Hao, L.; Hussein, A.; Young, P. Ti-6Al-4V triply periodic minimal surface structures for bone implants fabricated via selective laser melting. J. Mech. Behav. Biom. Mater. 2015, 51, 61–73.
  • Zhao G, Fu T, Li J. Study on Concave Direction Impact Performance of Similar Concave Hexagon Honeycomb Structure. Materials (Basel). 2023 Apr 21;16(8):3262. doi: 10.3390/ma16083262. PMID: 37110098; PMCID: PMC10147094
  • Zhao, G.; Fu, T. A Unit Compound Structure Design: Poisson’s Ratio Is Autonomously Adjustable from Negative to Positive. Materials 2023, 16, 1808. https:// doi.org/10.3390/ma16051808
  • Li, D.; Liao, W.; Dai, N.; Xie, Y.M. Comparison of Mechanical Properties and Energy Absorption of Sheet-Based and Strut-Based Gyroid Cellular Structures with Graded Densities. Materials2019, 12, 2183. https://doi.org/10.3390/ma12132183
  • Scherer, M.R.J. (2013). Double-Gyroid-Structured Functional Materials. Springer Theses. doi:10.1007/978-3-319-00354-2
  • Hailu, Y.M., Nazir, A., Hsu, CP. et al. Investigation of torsional properties of surface- and strut-based lattice structures manufactured using multiJet fusion technology. Int J Adv Manuf Technol 119, 5929–5945 (2022). https://doi.org/10.1007/s00170-022-08681-8
  • Monkova, K. Basics of determining the integral characteristics of planar and spatial formations, 1st edition, RISE Association, Prague - 2016, p.159, ISBN 978-80-87670-19-4.
  • Indurkar, P.P., Shaikeea, A., Xu, Z. et al. The coupled strength and toughness of interconnected and interpenetrating multi-material gyroids. MRS Bulletin 47, 461–473 (2022). https://doi.org/10.1557/s43577-021-00249-3
  • Feng, X., Burke, C. J., Zhuo, M., Guo, H., Yang, K., Reddy, A., … Thomas, E. L. (2019). Seeing mesoatomic distortions in soft-matter crystals of a double-gyroid block copolymer. Nature, 575(7781), 175–179. doi:10.1038/s41586-019-1706-1
  • Maskery, I., Aboulkhair, N. T., Aremu, A. O., Tuck, C. J., & Ashcroft, I. A. (2017). Compressive failure modes and energy absorption in additively manufactured double gyroid lattices. Additive Manufacturing, 16, 24–29. doi:10.1016/j.addma.2017.04.003
  • Aremu, I. Maskery, C. Tuck, I. Ashcroft, R. Wildman, and R. Hague, “A comparitive finite element study of cubic unit cells for selective laser melting,” in Solid Freeform Fabrication Symposium, 2014
  • Evsevleev, S.; et al. X-ray Computed Tomography Procedures to Quantitatively Characterize the Morphological Features of Triply Periodic Minimal Surface Structures. Materials 2021, 14, 3002. https://doi.org/10.3390/ma14113002
  • Khrapov D, Kozadayeva M, Manabaev K, Panin A, Sjöström W, Koptyug A, Mishurova T, Evsevleev S, Meinel D, Bruno G, Cheneler D, Surmenev R, Surmeneva M. Different Approaches for Manufacturing Ti-6Al-4V Alloy with Triply Periodic Minimal Surface Sheet-Based Structures by Electron Beam Melting. Materials (Basel). 2021 Aug 29;14(17):4912. doi: 10.3390/ma14174912. PMID: 34501001; PMCID: PMC8434612.
  • Yan, C., Hao, L., Hussein, A., & Young, P. (2015). Ti–6Al–4V triply periodic minimal surface structures for bone implants fabricated via selective laser melting. Journal of the Mechanical Behavior of Biomedical Materials, 51, 61–73. doi:10.1016/j.jmbbm.2015.06.024
  • Hosseini, E., & Popovich, V. A. (2019). A review of mechanical properties of additively manufactured Inconel 718. Additive Manufacturing, 30, 100877. doi:10.1016/j.addma.2019.100877
  • Nazir, A.; Ali, M.; Jeng, J.-Y. Investigation of Compression and Buckling Properties of a Novel Surface-Based Lattice Structure Manufactured Using Multi Jet Fusion Technology. Materials2021, 14, 2599. https://doi.org/10.3390/ma14102599
  • Madhusudan, S., Sarcar, M. M. M., & Rao, N. B. R. M. (2016). Mechanical properties of Aluminum-Copper(p) composite metallic materials. Journal of Applied Research and Technology, 14(5), 293–299. doi:10.1016/j.jart.2016.05.009
  • Singer, R., Ollick, A. M., & Elhadary, M. (2021). Effect of cross-head speed and temperature on the mechanical properties of polypropylene and glass fiber reinforced polypropylene pipes. Alexandria Engineering Journal, 60(6), 4947–4960. doi:10.1016/j.aej.2021.03.073 
  • Sood A, Ramarao S, Carounanidy U. Influence of different crosshead speeds on diametral tensile strength of a methacrylate based resin composite: An in-vitro study. J Conserv Dent. 2015 May-Jun;18(3):214-7. doi: 10.4103/0972-0707.157253. PMID: 26069407; PMCID: PMC4450527.
  • Alves, L. M., Nielsen, C. V., & Martins, P. A. F. (2011). Revisiting the Fundamentals and Capabilities of the Stack Compression Test. Experimental Mechanics, 51(9), 1565–1572. doi:10.1007/s11340-011-9480-5 
  1. Reviewer: Why double gyroid structure was made? What is the intended application and benefit from this double complexity?

The following text was added into the manuscript to better characterize double Gyroid structure and the reason of the research of its properties:

The mechanical properties of mono Gyroid structures (of different types - Sheet-Based and Strut-Based Gyroid [38]) made of different materials with different densities have been investigated in many studies. However, from the point of view of mechanical properties, the double Gyroid has remained in the background of scientific research until now, despite the fact that its other properties (especially optical, photonic, electrical, sound and others) are relatively well researched, an example of which is the comprehensive study of Sherer [39].

Since other types of properties of the double Gyroid look advantageous and promising for their application, the authors, therefore, decided to devote themselves to the study of this double cell structure Gyroid - Gyroid also within the research of the influence of the cross-head speed on its properties in comparison with the properties of a mono Gyroid.

  1. Reviewer: Force and displacement chart are shown in fig. 5. How about stress strain of these structure? Can authors calculate the stress strain from load deformation data and present stress strain graphs accordingly?

All the measured data were transferred and the stress was calculated from the load-displacement data. An example of stress-strain plots/dependences for both mono and double Gyroid structures with all investigated volume ratios has already been shown in the original manuscript in Figure 8.

  1. Reviewer: It is already well known that cross head speed does not significantly affect the lattice structure property, why is it one of the important parameters in this study. Authors can do a literature review and find out many papers which establish this parameter.

On the one hand, we agree with the Reviewer and acknowledge that the influence of the crosshead on the compressive properties of materials and porous structures has already been investigated in several studies as it is listed below, but on the other hand, we claim that the influence of the crosshead at combined structures, and namely at the double Gyroid, made of aluminium alloy by DMLS technology, has not yet been investigated. Therefore, the authors included this element in their research portfolio. Its result is the confirmation of the hypothesis based on other authors' research regarding the influence of the speed of the crosshead in the range of 1 to 100 mm/min on the mechanical properties in compression, but at the same time, the results pointed to other factors related to the double structure and the influence of the topology on its mechanical properties. The following References have been added into the manuscript: [40-43]

Madhusudan, S., Sarcar, M. M. M., & Rao, N. B. R. M. (2016). Mechanical properties of Aluminum-Copper(p) composite metallic materials. Journal of Applied Research and Technology, 14(5), 293–299. doi:10.1016/j.jart.2016.05.009

Singer, R., Ollick, A. M., & Elhadary, M. (2021). Effect of cross-head speed and temperature on the mechanical properties of polypropylene and glass fiber reinforced polypropylene pipes. Alexandria Engineering Journal, 60(6), 4947–4960. doi:10.1016/j.aej.2021.03.073 

Sood A, Ramarao S, Carounanidy U. Influence of different crosshead speeds on diametral tensile strength of a methacrylate based resin composite: An in-vitro study. J Conserv Dent. 2015 May-Jun;18(3):214-7. doi: 10.4103/0972-0707.157253. PMID: 26069407; PMCID: PMC4450527.

Alves, L. M., Nielsen, C. V., & Martins, P. A. F. (2011). Revisiting the Fundamentals and Capabilities of the Stack Compression Test. Experimental Mechanics, 51(9), 1565–1572. doi:10.1007/s11340-011-9480-5 

  1. Reviewer: What is the method of calculating area of cross section of gyroid structure shown in fig. 7? Software might not show an appropriate area. For area calculation, authors may review Investigation of torsional properties of surface- and strut-based lattice structures hailu et al. 

PTC Creo software uses differential equations to calculate all surface cross-sectional characteristics. The results of the calculations running in the background were analytically calculated and verified by the co-author of this manuscript and they were published in the book:

[52] Monkova, K. Basics of determining the integral characteristics of planar and spatial formations, 1st Edition, RISE Association, Prague - 2016, p.159, ISBN 978-80-87670-19-4.

The following Reference has been also added into the manuscript:

[14] Hailu, Y.M., Nazir, A., Hsu, CP. et al. Investigation of torsional properties of surface- and strut-based lattice structures manufactured using multiJet fusion technology. Int J Adv Manuf Technol 119, 5929–5945 (2022). https://doi.org/10.1007/s00170-022-08681-8

Reviewer 3 Report

1.       The authors mentioned TPMS structures and I would recommend a bit extended analysis of the literature data to provide enhanced novelty of the manuscript. For example, TPMS structures of Ti64 alloy have been printed via EBM by Koptyug et al, e.g. 10.3390/ma14113002; 10.3390/ma14174912

2.       Inconel 718 alloy is well described including its benefits, however, there are a lot of elements in particular potentially toxic as Ni. Thus, prospective applications can be limited.

3.       In respect with the mechanical properties of Gyroid + Gyroid samples compared with pure gyroid samples, provide some discussion on their benefits compared with available in literature alternatives.

4.       Alongside with the force-displacement curves, stress-strain curves are presented, which enable direct extraction of the most important mechanical properties such as Young moduli and compression strength. These values are presented; however, I would recommend a more detailed comparison with the literature data. In addition, provide statistical data that is the values should be presented as mean and standard deviations to reveal any statistically significant differences between the groups;

5.       The authors reported that Gyroid + Gyroid structure revealed deterioration of properties due to insufficient wall thickness, thus could we still expect similar or even better results compared with gyroid structures in case wall thicknesses are optimized.

Author Response

Dear Reviewer!

Thank You very much for Your valuable comments. We received them with great respect. We also would like to thank You for the possibility to make the improvements. We appreciate it very much.

All changes incorporated in the manuscript are highlighted in blue and responses to You are coloured in green.

  1. Reviewer: The authors mentioned TPMS structures and I would recommend a bit extended analysis of the literature data to provide enhanced novelty of the manuscript. For example, TPMS structures of Ti64 alloy have been printed via EBM by Koptyug et al, e.g. 3390/ma1411300210.3390/ma14174912

The literature data was expanded (please, see the authors' response to the Reviewer in point 3, as well as see the coloured text in the introductory part of the improved manuscript). The following references were included in the manuscript related TPMS:

Evsevleev, S.; et al. X-ray Computed Tomography Procedures to Quantitatively Characterize the Morphological Features of Triply Periodic Minimal Surface Structures. Materials 2021, 14, 3002. https://doi.org/10.3390/ma14113002

Khrapov D, Kozadayeva M, Manabaev K, Panin A, Sjöström W, Koptyug A, Mishurova T, Evsevleev S, Meinel D, Bruno G, Cheneler D, Surmenev R, Surmeneva M. Different Approaches for Manufacturing Ti-6Al-4V Alloy with Triply Periodic Minimal Surface Sheet-Based Structures by Electron Beam Melting. Materials (Basel). 2021 Aug 29;14(17):4912. doi: 10.3390/ma14174912. PMID: 34501001; PMCID: PMC8434612.

Yan, C., Hao, L., Hussein, A., & Young, P. (2015). Ti–6Al–4V triply periodic minimal surface structures for bone implants fabricated via selective laser melting. Journal of the Mechanical Behavior of Biomedical Materials, 51, 61–73. doi:10.1016/j.jmbbm.2015.06.024

  1. Reviewer: Inconel 718 alloy is well described including its benefits, however, there are a lot of elements in particular potentially toxic as Ni. Thus, prospective applications can be limited.

We agree with the reviewer, but on the other hand, due to the extraordinary combination of properties that this alloy is characterized by (we would like kindly ask the reviewer to see the improved material characteristic in the manuscript), there is still enough space for the application of the material and its practical use.

  1. Reviewer:In respect with the mechanical properties of Gyroid + Gyroid samples compared with pure gyroid samples, provide some discussion on their benefits compared with available in literature alternatives.

In addition to other research scattered throughout the text, the following studies dealing with the structure of the double gyroid were added to the manuscript:

One form of the gyroid lattice known as the double Gyroid was recently identified as having high stiffness and low maximum von Mises stress compared to a variety of other cell types making it particularly suitable for use in lightweight components. Furthermore, Aremu et al. [34] noted that the double Gyroid lattice, unlike several other lattice types, possesses axisymmetric stiffness, again making it a good candidate for applications where the exact nature and direction of the loads are not fully known or if they are subject to large uncertainties.

The energy absorption of as-built and heat-treated double Gyroid lattice structures was studied by Maskery et al. [35] Their results showed total energy absorbed by heat-treated double Gyroid lattices up to 50% was close to three times the energy absorbed by the examined single BCC lattices, which had the same volume ratio.

Scientists and engineers are interested in gyroids because of the way they interact with both light and sound waves, promising nanoscale materials with novel properties. The gyroid's form dictates how and even whether a wave will pass through to the other side. In that way, the material can be invisible to some waves, or a reflector of other wavelengths. Remarkably, the chemical combination of polydimethylsiloxane (PDMS) and polystyrene, initially dissolved in a solution, self-assembles into a double gyroid, with two distinct PDMS networks dancing around each other without ever touching. A double gyroid can be even more tunable, as distinct materials making each network could affect signals differently. All this is predicated on the unit cell structure being perfect cubes. [36]

The properties of double Gyroid, that the constituent interpenetrating networks are single gyroids of opposite chirality, were studied also by Indurkar et. al. They stated that it is straightforward to induce connectivity between the two networks by translating one with respect to the other. The interconnected and interpenetrating gyroids of opposite chirality are a bending-dominated topology. The increase in nodal connectivity by inducing interconnectivity within them is insufficient to switch their behaviour to stretching-dominated. [37]

The mechanical properties of mono Gyroid structures (of different types - Sheet-Based and Strut-Based Gyroid [38]) made of different materials with different densities have been investigated in many studies. However, from the point of view of mechanical properties, the double Gyroid has remained in the background of scientific research until now, despite the fact that its other properties (especially optical, photonic, electrical, sound and others) are relatively well researched, an example of which is the comprehensive study of Sherer [39].

Since other types of properties of the double Gyroid look advantageous and promising for their application, the authors, therefore, decided to devote themselves to the study of this double cell structure Gyroid - Gyroid also within the research of the influence of the cross-head speed on its properties in comparison with the properties of a mono Gyroid.

  1. Reviewer: Alongside with the force-displacement curves, stress-strain curves are presented, which enable direct extraction of the most important mechanical properties such as Young moduli and compression strength. These values are presented; however, I would recommend a more detailed comparison with the literature data. In addition, provide statistical data that is the values should be presented as mean and standard deviations to reveal any statistically significant differences between the groups;

The literature data was expanded as it was mentioned in the point 1) within the authors´ responses to the Reviewer´s comments.

As we mentioned in the original manuscript, due to the high production cost of Inconel 718 samples and the financial capabilities of the authors' workplaces, only one piece per order of each type of sample was produced. Each sample represents a separate group characterized by different boundary conditions of production and testing; therefore, statistical processing was not done.

  1. Reviewer: The authors reported that Gyroid + Gyroid structure revealed deterioration of properties due to insufficient wall thickness, thus could we still expect similar or even better results compared with gyroid structures in case wall thicknesses are optimized.

We added to the improved manuscript, as part of the discussion under Fig. 10, other possible factors that could influence the lower mechanical properties of the double Gyroid structure:

One of the other causes of worse mechanical properties of double Gyroid structure is the "crossing of the walls", which can affect not only the technological conditions of production (e.g. laser beam guidance strategy that thereby can influence the microstructure of the material during the passing of the laser in a new layer) but at the same time, it is probably necessary to take into account that at the crossing of the walls, almost sharp inner edges are created, which tend to behave in the sense of a notch, which presumably also subsequently affects the mechanical properties of the double gyroid.

Considering the above, it is not possible to make definite conclusions, but at least in the case of optimization of the wall thickness and the use of topology with rounded internal corners in the cross-sections of the structure, it is possible to assume that there would be an improvement in the mechanical properties. The use of mathematical apparatus for topology optimization will be another challenge for scientists dealing with the properties of such types of structures.

Round 2

Reviewer 1 Report

The updated version of the manuscript has improved the quality of the presentation of the study. 

Reference 15 and 16 should be corrected.

15. Zhao G, Fu T, Li J. Study on Concave Direction Impact Performance of Similar Concave Hexagon Honeycomb Structure[J]. Materials, 2023, 16(8): 3262.

16. Zhao G, Fu T. A unit compound structure design: poisson’s ratio is autonomously adjustable from negative to positive[J]. Materials, 2023, 16(5): 1808.

Author Response

Dear Reviewer!

Thank You very much for Your valuable comments. We received them with great respect. We also would like to thank You for the possibility to make the improvements. We appreciate it very much.

All changes in the manuscript and responses are highlighted in green.

Reviewer: Reference 15 and 16 should be corrected.

The Refences 15 and 16 were corrected according to the Reviewer’s recommendation.

Reviewer 2 Report

Following comments has not been addressed or answered properly.

1. "Thus, reviewer would like to emphasize to consider some more parameter and their effect on the studied lattices in order to make this study more valuable for the readers and peers. Authors can consider studying the energy absorption behavior of the studied structures as well. " This is an important and major comment since the results shows that cross head speed doesn't effect the mechanical properties significantly. Therefore, it was advised to explore another aspect of these structures.

2. "Since other types of properties of the double Gyroid look advantageous and promising for their application, the authors, therefore, decided to devote themselves to the study of this double cell structure Gyroid. " What other properties? Please specifically mention the application and significance of double gyroid structures. 

OKAY.

Author Response

Dear Reviewer!

Thank You very much for Your valuable comments. We received them with great respect. We also would like to thank You for the possibility to make the improvements. We appreciate it very much.

All changes in the manuscript and responses are highlighted in green.

  1. Reviewer: "Thus, reviewer would like to emphasize to consider some more parameter and their effect on the studied lattices in order to make this study more valuable for the readers and peers. Authors can consider studying the energy absorption behavior of the studied structures as well. " This is an important and major comment since the results shows that cross head speed doesn't effect the mechanical properties significantly. Therefore, it was advised to explore another aspect of these structures.

To make the study more valuable for readers and peers, the authors evaluated the energy needed for absorption by a structure up to 60 % of strain. The following texts and graphs were added into the manuscript:

2.3. Evaluation of energy absorption

The behaviour of materials and structures plays a key role in the safety of product under load. Energy absorption refers to the process of dissipating input energy from an external load through plastic deformation or fracture and it corresponds to the action of the force on the displacement. [52] A force F acting through an elementary displacement dL performance elementary work FdL. A stress s = F/A acting through a strain increment de = dL/L does work

dW = FdL/AL = s de                                                                                                (2)

where A is cross-sectional area on which force acts. If the stress is acting on an elastic material, this work is stored as elastic energy. The work done per unit volume as the stress is raised from zero to a final value s* is the area under the stress-strain curve:

(3)

This is the energy W (J) that is stored, per unit volume, in an elastically strained material. The energy is released when the stress is relaxed.

Within the research, the measured force-displacement curves were used for calculation of energy absorptions of the individual samples (differ in the volume ratio Vr, type of structure G and GG, as well as in crosshead speed). Although the compression tests were carried out without splintering fracture up to approx. 80 % compressive strain (except for the Diamond build which was slightly compressed due to overload), final strains varied. In order to make the results comparable, the authors decided to use a strain of 60% as a criterion for evaluating energy absorption.

Since the plateau regions were not smooth, it was difficult to describe the force and displacement dependences using equations. This was the reason why the authors decided to use the discreditation method to calculate the energy instead of differentiation. The principle (Fig. 5) consists in dividing a large area into the sum of small areas corresponding to the area of the rectangle with sizes Δa and height f(a), where Δa is the increment of the value on the x-axis and f(a) is the functional value corresponding to the variable a, which in the interval Δa and hardly changes. [53] Software MS Excel was used for the area (i.e., energy absorption) calculation.

Figure 5. The principle of discreditation at the calculation of energy absorption

  1. Monkova, K.; Monka, P.P.; Žaludek, M.; Beňo, P.; Hricová, R.; Šmeringaiová, A. Experimental Study of the Bending Behaviour of the Neovius Porous Structure Made Additively from Aluminium Alloy. Aerospace 2023, 10, 361. https://doi.org/10.3390/aerospace10040361
  2. Monkova, K. Basics of determining the integral characteristics of planar and spatial formations, 1st edition, RISE Association, Prague - 2016, p.159, ISBN 978-80-87670-19-4.

Results and discussions …

Since the effect of the crosshead speed on the compressive properties of the samples hasn´t been observed, the authors decided to look at the samples´ behavior also from the energy absorption point of view. Based on the methodology described in the section 2.3., the dependencies of the energy absorption at 60 % strain on both crosshead speed and volume ratio were plotted separately for Gyroid and Gyroid + Gyroid structures. They are presented in Fig. 12.

(a)

(b)

Figure 12. Energy absorption at 60 % of strain, a) Gyroid; b) Gyroid + Gyroid

It is visible from the graphs above, that also in this case, the crosshead speed hasn´t influenced amount of the energy, while with volume ration the amount of the energy has increased at Gyroid in almost a linear way and at Gyroid + Gyroid slightly in exponential way. Comparing the maximal values, it can be said that the slightly higher results achieved Gyroid (60.12 kJ), while at the Gyroid + Gyroid structure it was 59,55 kJ, both at the crosshead speed 10 mm/min.

  1. Reviewer: "Since other types of properties of the double Gyroid look advantageous and promising for their application, the authors, therefore, decided to devote themselves to the study of this double cell structure Gyroid. " What other properties? Please specifically mention the application and significance of double gyroid structures. 

The following text was added into the manuscript to show the promising properties of double Gyroid structure:

Although the results did not show better compressive properties of the Gyroid + Gyroid structure, the porous structure of the Gyroid itself is characterized by a large surface area, which is very useful for heat conduction and light absorption (or accumulation of light energy). This function is doubled by the Gyroid + Gyroid structure, so the use of the double gyroid structure is promising for heat exchangers and batteries, increasing their capacity.
